# Temporal Interpolation via Motion Field Prediction

**Lin Zhang\*, Neerav Karani\*, Christine Tanner, Ender Konukoglu**
Computer Vision Laboratory, ETH Zurich
`nkarani@vision.ee.ethz.ch`

## Abstract

Navigated 2D multi-slice dynamic Magnetic Resonance (MR) imaging enables high contrast 4D MR imaging during free breathing and provides in-vivo observations for treatment planning and guidance. Navigator slices are vital for retrospective stacking of 2D data slices in this method. However, they also prolong the acquisition sessions. Temporal interpolation of navigator slices can be used to reduce the number of navigator acquisitions without degrading specificity in stacking. In this work, we propose a convolutional neural network (CNN) based method for temporal interpolation via motion field prediction. The proposed formulation incorporates the prior knowledge that a motion field underlies changes in the image intensities over time. Previous approaches that interpolate directly in the intensity space are prone to produce blurry images or even remove structures in the images. Our method avoids such problems and faithfully preserves the information in the image. Further, an important advantage of our formulation is that it provides an unsupervised estimation of bi-directional motion fields. We show that these motion fields can be used to halve the number of registrations required during 4D reconstruction, thus substantially reducing the reconstruction time.

## 1 Introduction

Quantification of breathing-induced motion of anatomical structures is an important component in image-guided therapy applications, such as planning and guiding radiotherapy [1] and high intensity focused ultrasound therapy [2]. The main source for observing and quantifying long term motion patterns is dynamic volumetric MR imaging (4D-MRI) [3]. A particular type of 4D MRI, *navigated 2D multi-slice imaging*, is especially useful as it is acquired during free-breathing and can capture irregular breathing patterns, which require long and uninterrupted observations. In this technique, navigator slices $\mathbf{N}_t$ (at same anatomical location) and data slices $\mathbf{D}^p$ (at different locations $p$ to cover the volume of interest) are alternately acquired. Data slices enclosed by navigators which show the most similar organ position are retrospectively stacked together to create a 3D MRI for each navigator. 4D reconstruction without navigators has been proposed by using external breathing signal [4] or consistency between adjacent data slices [5]. However, navigators enable continuous organ motion quantification, which might not be externally measurable (e.g. drift of the liver [3]) or hard to accurately estimate from the data slices, and hence potentially provide superior reconstructions.

Although useful, navigators prolong the acquisition time. Reducing the number of navigator acquisitions without sacrificing accuracy in stacking of data slices can reduce total acquisition time or improve through-plane resolution if the saved time is used to acquire more data slices. This motivates the idea of acquiring fewer navigators and temporally interpolating these to predict the missing ones.

With this motivation, [6] proposed a convolutional neural network (CNN) based approach for temporal interpolation of navigators. Their CNN takes as inputs a fixed number of acquired images and learns to predict the missing images directly in the intensity space. This approach, which we call the *Simple Convolutional Interpolation Network (SCIN)*, is a 'black-box' formulation that does not incorporate

1st Conference on Medical Imaging with Deep Learning (MIDL 2018), Amsterdam, The Netherlands.
\*Both authors contributed equally to this manuscript.

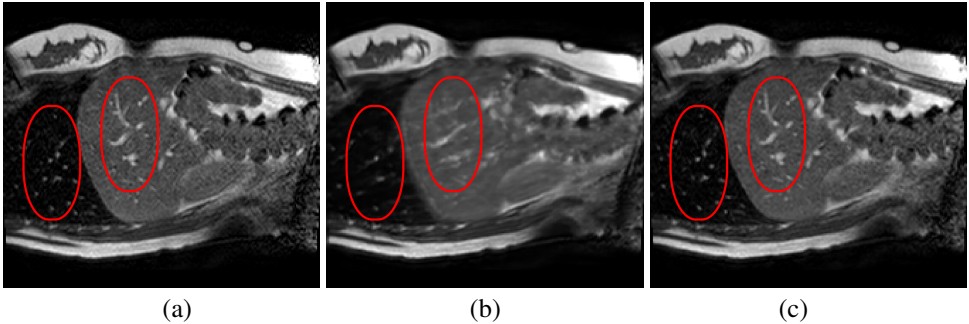

(a)            (b)            (c)

Figure 1: (a) Ground truth and (b,c) interpolated images from (b) baseline (SCIN) and (c) proposed method (MFIN). The image interpolated via SCIN is heavily blurred and misses several lung and liver structures, while the proposed method is able to preserve the details in the ground truth image.

any prior information about the interpolation process. Image prediction is guided only by the cost function used to optimize the network parameters. The issue with this is that it is unclear whether the image similarity measures that are generally used as cost functions suffice to ensure fidelity of the generated images to the original images. Indeed, Fig. 1b shows a case where an image interpolated using SCIN is quite blurry and misses several liver and lung structures present in the original image.

In this article, we propose an interpolation method that incorporates the prior knowledge that a motion field underlies the difference between images acquired at different times. We note that in scenarios where an image sequence captures anatomical structures in motion (without induced contrast changes), the content of the images remains largely unchanged over time and issues such as occlusion are not pertinent. Further, if the principal direction of motion is in the plane of the 2D images, the chances of structures going out of the image or new structures coming into the image due to off-plane motion may be minimal. Under these assumptions, each image can be viewed as a spatially transformed version of its temporal neighbours. This observation leads us to incorporate motion field prediction as an intermediate step for the interpolation problem, which removes the ability of the CNN to directly change image intensities and enables the regularization of the predicted motion fields. We hypothesize that this formulation makes changes in image structure unlikely, leading to more plausible predicted images. We train a CNN to take as input several acquired navigators and predict the motion fields between the image to be interpolated and its known two neighbours. Any of these two motion fields can then be used to wrap the corresponding known neighbouring image to obtain the missing image. We call this network the *Motion Field Interpolation Network* (MFIN). MFIN is trained end-to-end using only navigator images, without ground truth motion fields. Indeed, an important advantage of our interpolation formulation is that it provides an unsupervised estimation of the motion fields. In the particular setting of navigated multi-slice imaging, each navigator has to be registered to a reference image. The motion fields obtained by our interpolation framework can be used to halve the number of these registrations, substantially reducing the computational effort of 4D reconstruction.

## 2   Related Work

Temporal image interpolation in the medical imaging context has been mainly suggested for ultrasound imaging [7–9]. These methods explicitly track pixel-wise correspondences between neighbouring images via optical flow estimation or non-linear registration. An advantage of such methods is that they often estimate the underlying motion fields as part of the interpolation. Yet, they often make simplistic assumptions regarding the shape and dynamics of the motion trajectory such as linear, constant velocity. With the surge of learning-based methods, end-to-end learning-based solutions directly predict in-between images given surrounding ones, skipping the motion field estimation completely. In this line, [6] proposed the aforementioned SCIN for interpolating navigators for 4D-MRI reconstruction. In computer vision, variants of CNNs have been suggested for interpolation [10] and video frame prediction [11–13]. A common feature between these methods is that they train their networks to directly predict the missing image in the intensity space.

Prediction of image intensities from scratch may be difficult, leading to blurry results, or even distortion of image structures. To tackle this, some works have suggested using content of the known

images. [14] propose to use a variational auto-encoder (VAE) [15] to learn bidirectional motion fields between two known images. Then, without explicit training for interpolation, motion fields from the known images to an in-between image are predicted by linear interpolation in the latent space of the VAE. This approach produces sharp interpolated facial expression images, but it is unclear whether it would be able to faithfully interpolate breathing motion patterns in abdominal navigators. In [16], a CNN directly predicts bi-directional motion fields between the known images, which are further refined using another CNN to account for occlusions and then combined to produce the interpolated image. In [17], interpolation is formulated as a local convolution over the known images and a CNN predicts a convolutional kernel for each pixel of the interpolated image. Although these methods incorporate prior information about the image content into the interpolation framework, they do not readily provide the motion fields from the known images to the final predicted image. Note that in both [14] and [16], bi-directional motion fields are combined to obtain the interpolated image, as the emphasis is on dealing with interpolation scenarios including occlusions. Such combination renders the methods unable to provide the final motion field between the predicted and the known images.

The underlying motion estimation problem has been separately studied, either requiring ground truth flow fields for training [18–20] or in an unsupervised fashion by relying on reconstruction of warped images using predicted flow fields [21, 22]. The relationship between the problems of interpolation and motion field estimation has been exploited in [10] to find dense correspondences between input images via saliency maps [23] of an interpolation CNN. Instead of this indirect approach of using interpolation to find motion fields, MFIN takes the direct route and interpolates by first estimating the underlying motion.

Another related problem is that of image registration. CNNs have been proposed for learning image registration using known ground truth motion [24] or gold-standard registration results [25], or in an unsupervised manner within an optimization framework [26]. While the registration problem is one of motion field estimation between two known images, the interpolation problem is to predict missing images and here we are additionally interested in estimating the underlying motion field

## 3  Method

We consider the scenario where the temporal resolution of the acquired navigator sequence is sought to be doubled. That is, missing navigators $N_4$, $N_6$, $N_8$, etc. are to be interpolated using the acquired $N_1$, $N_3$, $N_5$, $N_7$, etc.. Following [6], where temporal context beyond immediate neighbours has been shown to be important for dealing with non-linear motion, we provide 2 images each from the past and the future as inputs to MFIN. Thus, in order to interpolate $N_t$, the inputs to the network are $N_{t-3}$, $N_{t-1}$, $N_{t+1}$ and $N_{t+3}$. The general architecture of MFIN is shown in Fig. 2. The inputs pass through shared convolutional layers before diverging into 2 sub-networks. Each sub-network predicts the motion field from the image to be interpolated, $N_t$, to one of its neighbours ($N_{t-1}$ or $N_{t+1}$). The motion field predicted by each sub-network ($\mathbf{F}_{t \to t-1}$ or $\mathbf{F}_{t \to t+1}$) is used to warp the corresponding neighbouring image using bilinear interpolation to predict $N_t$ independently. The warping is incorporated into the network and works as follows. The intensity of each pixel in the image to be interpolated is obtained via spatial bilinear interpolation in the neighbouring image ($N_{t-1}$ or $N_{t+1}$) around the location pointed by the corresponding predicted motion field ($\mathbf{F}_{t \to t-1}$ or $\mathbf{F}_{t \to t+1}$). The loss function used to optimize the network parameters (discussed in section 3.1) is defined to measure the similarity of the interpolated and the ground truth images, thus not requiring ground truth motion fields. While either one of the two sub-networks is sufficient for interpolation, we still predict the displacement fields in both directions in view of potential inductive bias promoted by multi-task learning [27]. Additionally, in an extension to our base model (described in Sec.3.2), we utilize the bidirectional motion fields to enforce a cyclic consistency constraint.

### 3.1  Loss Functions

The loss function for MFIN, shown in Eq. 1, consists of a reconstruction loss term ($L_{recon}$) and a regularization term ($L_{reg}$). $L_{recon}$ (Eq. 2) is the sum of the reconstruction errors from the two sub-networks, where $N'_{t,s}$ denotes the prediction for image $N_t$ be warping the image $N_s$ according to the estimated motion field $\mathbf{F}_{t \to s}$, i.e. defined by displacement vectors defined at pixel locations in $N_t$ pointing to $N_s$. The form of the reconstruction loss must capture the desired notion of image similarity between the predicted and the ground truth image. The mean-squared-error in intensity

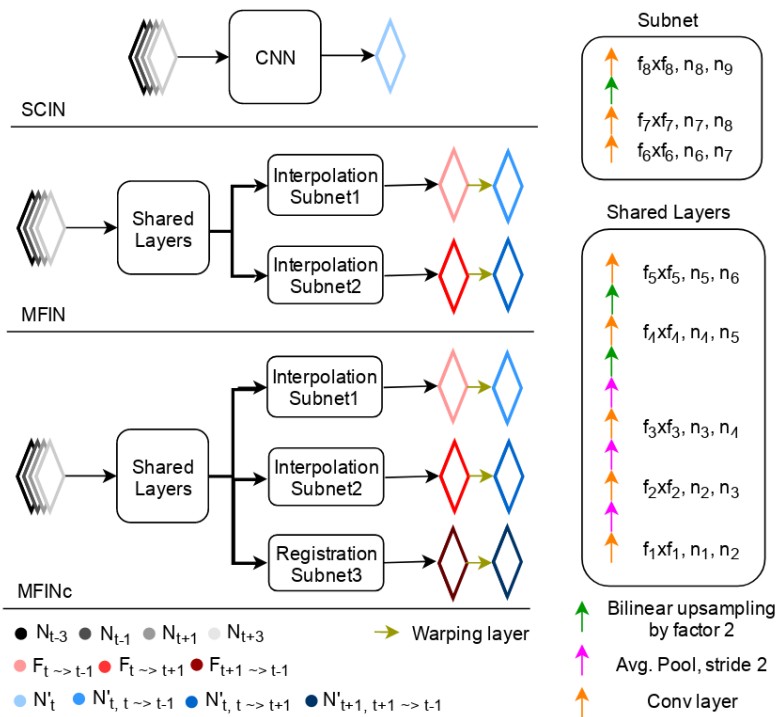

Figure 2: Architectures of the Simple Convolutional Interpolation Network [6] (SCIN), the proposed Motion Field Interpolation Network (MFIN) and its extension for incorporating cyclic consistency (MFINc). For the convolutional layers, $f_i$, $n_i$ and $n_{i+1}$ indicate the filter size and number of input and output channels respectively in the $i^{th}$ layer.

space, which is generally used as the reconstruction loss, might not be robust to intensity scaling. Such scaling might be relevant due to saturation effects in the employed interleaved MR acquisition, where the navigator image is saturated when the preceding data slice is at similar location. Since the MFIN formulation simply moves the pixel-intensities from one of the known images to obtain the missing image, it cannot account for such effects. Importantly, such intensity differences are not crucial for the application of 4D reconstruction as long as the estimated motion fields are accurate. With this motivation, we investigate the use of structural similarity index (SSIM) [28] as a loss function in addition to the generally used $L_2$ intensity loss. The SSIM between two image patches $x$ and $y$ is defined as in Eq 4, where $\mu_x$, $\mu_y$ are patch means, $\sigma_x^2$, $\sigma_y^2$ are patch variances, $\sigma_{xy}$ is the covariance of the two patches and $c1, c2$ are constants. The SSIM for the entire image is taken as the mean of the patch similarities. It takes into account local correlation between image patches and therefore, may be more robust to the aforementioned intensity scalings. It has also been shown to preserve low-level structure [29]. Finally, it is differentiable and can thus be readily integrated into gradient based optimization. As SSIM measures image similarity instead of image dissimilarity, $L_{recon}$ is maximized when SSIM is employed as the loss function. For $L_{reg}$, we employ total variation regularization (Eq. 3) to promote smoothness in the predicted motion fields, while allowing for sharp gradients to cope with sliding interfaces.

$$L_{total,MFIN} = L_{recon,MFIN} + \lambda_1 L_{reg,MFIN} \tag{1}$$

$$L_{recon,MFIN} = L(N'_{t,t-1}, N_t) + L(N'_{t,t+1}, N_t) \tag{2}$$

$$L_{reg,MFIN} = ||\nabla \mathbf{F}_{t \to t-1}||_1 + ||\nabla \mathbf{F}_{t \to t+1}||_1 \tag{3}$$

$$SSIM(x,y) = \frac{(2\mu_x\mu_y + c1)(2\sigma_{xy} + c2)}{(\mu_x^2 + \mu_y^2 + c1)(\sigma_x^2 + \sigma_y^2 + c2)} \tag{4}$$

## 3.2 Cycle Consistency

Cycle consistency has been shown to be an effective regularizer in registration problems [30, 31] as well as in other contexts such as image generation in deep neural networks [32]. The MFIN architecture can be readily extended to include such cyclic consistency between estimated motion fields. For this, we add another sub-network to the MFIN for estimating the motion field between the two, always known, neighbours of the image to be interpolated, i.e. $\mathbf{F}_{t+1 \to t-1}$. We denote the extended network by MFINc and optimize for the registration and two interpolation tasks jointly by minimizing the cost function in Eq. 5. The reconstruction (Eq. 6) and regularization (Eq. 7) terms in the loss function are extended to include the extra sub-network. Finally, Eq. 8 shows the cycle consistency loss term, where $\odot$ denotes a pixel-wise composition of transformations achieved via bilinear interpolation of the motion field. We hypothesize that the registration sub-network has an easier task in that it has to estimate the motion field between two known images, while the interpolation sub-networks have to predict the missing image in addition to estimating the corresponding motion fields. Thus, $\mathbf{F}_{t+1 \to t-1}$ may be more accurate than $\mathbf{F}_{t \to t-1}$, $\mathbf{F}_{t \to t+1}$ and enforcing cycle consistency may help to correct errors in the latter two.

$$L_{total,MFINc} = L_{recon,MFINc} + \lambda_1 L_{reg,MFINc} + \lambda_2 L_{cycle,MFINc} \tag{5}$$

$$L_{recon,MFINc} = L(N'_{t,t-1}, N_t) + L(N'_{t,t+1}, N_t) + L(N'_{t-1,t+1}, N_{t-1}) \tag{6}$$

$$L_{reg,MFINc} = ||\nabla \mathbf{F}_{t \to t-1}||_1 + ||\nabla \mathbf{F}_{t \to t+1}||_1 + ||\nabla \mathbf{F}_{t+1 \to t-1}||_1 \tag{7}$$

$$L_{cycle,MFINc} = ||(\mathbf{F}_{t+1 \to t-1} \odot \mathbf{F}_{t \to t+1}) - \mathbf{F}_{t \to t-1}||_2 \tag{8}$$

# 4 Experiments and Results

## 4.1 Dataset

We carry out our experiments on 2D navigator images from an abdominal 4D MRI dataset consisting of 14 subjects. The interleaved acquisition of navigator and data slices [3] was done on a 1.5T Philips Achieva scanner using a 4-channel cardiac array coil, a balanced steady-state free precession sequence, SENSE factor 1.7, $70^o$ flip angle, 3.1 ms TR, and 1.5 ms TE. The images have a spatial resolution of $1.33 \times 1.33$ mm$^2$, slice thickness of 5mm and temporal resolution of 2.4-3.1 Hz. For each subject, there are between 4000 and 6000 navigators, acquired in several blocks of 7 to 9 min and with 5 min resting periods in between. Expert-annotated landmarks for two liver vessels per image were available for 10% randomly selected images for 7 out of the 14 subjects. We train on the remaining 7 subjects and use the 7 subjects with expert annotations as test subjects, so that the accuracy of the predicted motion fields could be evaluated as described in Sec. 4.3.

## 4.2 Implementation details

We implement MFIN and MFINc (Fig. 2) as networks with an encoder-decoder like structure. Both networks consist of an initial block of shared layers, followed by 2 and 3 separate sub-networks for MFIN and MFINc respectively. The shared layers include the entire encoder / contracting path as well as two upscaling layers from the decoder. Each sub-network consists of two convolutional layers, followed by a bilinear upsampling and then a final convolutional layer to obtain a motion field. There is no activation function at the end of the last convolutional layer to allow for both positive and negative flow values. All other convolutional layers are following by a ReLU activation function. Finally, a warping layer maps the predicted motion field and the corresponding known image to the interpolated image. The filter sizes and number of feature maps are empirically set to ($f_1$, $f_2$, ..., $f_8$) = (7,5,3,3,3,3,3,3) and ($n_2$, $n_2$, ..., $n_8$) = (16,32,64,64,32,32,16). For interpolating a navigator at any time point, two known navigators from the past and the future are provided as inputs. The output of each sub-network before the warping layer is a 2D flow vector for each pixel, i.e. ($n_1$, $n_9$) = (4, 2). Following [6], we set the batch size to 64 and use the Adam optimizer [33] with a learning rate of 1e-4. The only pre-processing step is block-wise linear normalization of the images to their 2 to 98 %tile range. Following [28], the hyperparameters of the SSIM loss are set as $c1 = 0.0001$, $c2 = 0.0009$ and patch size of 11 x 11. The weights for the regularizers in the loss functions are

empirically set to $\lambda1 = 0.001$ and $\lambda2 = 0.0005$ while using the $L_2$ loss for $\mathrm{L}_{recon}$ and $\lambda1 = 0.1$ and $\lambda2 = 0.05$ while using the SSIM loss for $\mathrm{L}_{recon}$.

## 4.3 Evaluation

The choice of appropriate evaluation metrics is crucial to correctly compare competing solutions. Here, we list several metrics and discuss their suitability for evaluating interpolation in the setting of navigator slices for 4D MR reconstruction.

1. **Root-mean-squared-error (RMSE)**: RMSE or mean-absolute-error are generally used for evaluation in regression problems. However, they might not be well-suited for measuring image fidelity due to the following reasons. Firstly, they are measured pixel-wise and hence, do not encode structural information. Further, these metrics are not robust to intensity scaling. In the case of interpolation of navigators, we are only interested in the correct location of the structures in the image. Thus, a shift in the average intensity should not be punished. This might be relevant in cases when the navigator is affected by saturation, as in acquisitions with interleaved navigator and data slices. Finally, these metrics are sensitive to noise and noise is inherently present in MR images. The $L_2$ distance between a given ground truth image and a smoothened (denoised) interpolated image would be smaller than the case where the interpolated image had different noise values that the ground truth image.

2. **Structural Similarity Index (SSIM)**: SSIM encodes local correlations between image patches along with their intensity similarity. Thus, it may be expected to be more robust to intensity scaling and image distortions than RMSE. However, it is not robust to noise [34] and also does not directly capture any important clinically relevant aspects like correct organ position or accurate motion estimation.

3. **Residual motion (ResMot)**: We register the interpolated image to the ground truth image via a gold standard (gs) image registration algorithm (linearly interpolated grid of control points, optimized for local correlation coefficient, total variation regularization) [35], thus obtaining an error motion field $\mathbf{F}^{\mathrm{gs}}_{t \to \hat{t}}$. The mean magnitude of this motion field could be relevant for 4D reconstruction as it measures the mismatch in organ positions.

4. **Error in motion to a reference image (RefMotErrIm)**: For 4D reconstruction, each navigator image is registered to a reference image to estimate the position of the structure of interest in the navigator. To measure the error introduced in this step due to interpolation, we compute the difference of motion fields obtained via GS registration between the reference image and either (i) an interpolated navigator or (ii) a true navigator, thus obtaining flow fields $\mathbf{F}^{\mathrm{gs}}_{t \to ref}$ or $\mathbf{F}^{\mathrm{gs}}_{\hat{t} \to ref}$ respectively. From these flow fields, we compute two evaluation measures: the mean difference in their magnitudes over the entire image (RefMotErrIm) or only over the structure of interest, the liver in this application (RefMotErrImLiver).

The interpolation formulation in MFIN provides an unsupervised estimation of the motion fields between the interpolated image and its neighbours. We use the following measures for evaluating the accuracy for these motion fields.

5. **Using the estimated motion fields for determining positions of interpolated images (RefMotErrFl)**: As mentioned before, the crucial step in 4D reconstruction is to estimate the position of the structure of interest in each navigator. This is usually done by registering each navigator to a reference image and is the most time consuming step in the reconstruction based on [3]. Since the interpolation of a navigator $\mathrm{N}_t$ provides the motion field $\mathbf{F}_{t \to t+1}$, we can use it to reduce the number of navigator registrations by half. This can be achieved by inverting the predicted motion field $\mathbf{F}_{t \to t+1}$ to get $\mathbf{F}_{t+1 \to t}$ and then composing it with $\mathbf{F}^{\mathrm{gs}}_{ref \to t+1}$ (obtained by registering $\mathrm{N}_{t+1}$ to the reference image) to estimate $\mathbf{F}^{\mathrm{gs}}_{ref \to t}$. The error in the estimation ( $\mathbf{F}_{t+1 \to t} \circ \mathbf{F}^{\mathrm{gs}}_{ref \to t+1}$ - $\mathbf{F}^{\mathrm{gs}}_{ref \to t}$) serves as a measure of the accuracy of the predicted motion field $\mathbf{F}_{t \to t+1}$. As with measure [4], [5] can also be computed either over the entire image (RefMotErrFl) or only over the structure on interest (RefMotErrFlLiver).

6. **Landmark error (LandmarkErr)**: Another method to evaluate the accuracy of the motion fields is to compute the landmark errors for the cases where we have expert annotations on consecutive navigators.

Table 1: Quantitative results. %ile refers to 5 percentile values for SSIM and 95 percentile otherwise.

| Evaluation Metric | SCIN-$L_2$ mean | %ile | MFIN-$L_2$ mean | %ile | MFINc-$L_2$ mean | %ile | SCIN-SSIM mean | %ile | MFIN-SSIM mean | %ile | MFINc-SSIM mean | %ile |
|---|---|---|---|---|---|---|---|---|---|---|---|---|
| RMSE | **8.78** | **11.83** | 10.35 | 13.90 | 10.23 | 13.74 | 9.05 | 12.03 | 10.30 | 14.09 | 10.28 | 14.06 |
| SSIM [%] | 82.08 | 77.15 | 79.55 | 74.35 | 79.61 | 74.53 | **82.72** | **78.01** | 79.81 | 74.67 | 79.87 | 74.78 |
| ResMot [mm] | **0.30** | **0.55** | 0.37 | 0.67 | 0.36 | 0.67 | **0.30** | 0.58 | 0.37 | 0.68 | 0.37 | 0.67 |
| RefMotErrIm | **0.53** | **0.98** | 0.58 | 1.05 | 0.57 | 1.04 | 0.54 | 1.00 | 0.59 | 1.08 | 0.58 | 1.05 |
| RefMotErrImLiver | 0.70 | 1.43 | 0.72 | 1.44 | 0.70 | 1.42 | **0.66** | 1.37 | 0.70 | 1.39 | 0.68 | **1.35** |
| RefMotErrFl [mm] | - | - | 0.83 | 1.64 | 0.84 | 1.64 | - | - | 0.84 | 1.66 | **0.82** | **1.63** |
| RefMotErrFlLiver | - | - | 0.83 | 1.63 | 0.83 | 1.66 | - | - | 0.80 | 1.57 | **0.78** | **1.53** |
| LandmarkErr [mm] | - | - | 0.98 | 1.88 | 0.93 | 1.97 | - | - | 0.93 | 1.85 | **0.92** | **1.81** |

## 4.4 Experiments and Results

We trained three networks: SCIN, MFIN and MFINc. Each network was trained separately with two loss functions for $L_{recon}$: the $L_2$ loss and SSIM. Table 1 summarizes quantitative results in terms of the evaluation metrics discussed in Sec. 4.3. In terms of RMSE and SSIM, SCIN performs better than both MFIN and MFINc. However, as discussed in Sec. 4.3, these metrics might not be appropriate for measuring interpolation performance.

Among the registration-based evaluation measures, SCIN performs better in terms of ResMot. Yet this measure might to some extend be artifically reduced for SCIN, as its blurring and denoising property is likely to reduce gradients of the image similarity measure optimized during registration. RefMotErrImLiver is the most relevant evaluation measure for the application of reconstruction of 4D MRIs. With regard to this measure, the difference in mean performance between SCIN-SSIM and MFINc-SSIM is 0.02mm. To test whether this difference in performance affects the reconstruction, we computed the error incurred in the data slice sorting that follows the navigator position determination step in 4D reconstruction. In the case where all ground truth navigators are used, the discrepancy between the navigator and the closest data slice is, on average, 1.48mm. This is much larger than the difference in RefMotErrImLiver between SCIN and MFIN or MFINc. We thus infer that the increase in RefMotErrImLiver for MFINc or even for MFIN as compared to SCIN may not affect the reconstruction. Note that RefMotErrImLiver is higher than RefMotErrIm because the motion magnitude to the reference is higher in the liver (mean 5.13, 95% 11.81 mm) than for whole image (mean 3.40, 95% 8.21 mm).

Qualitative results are shown in Fig. 4. We observe no large qualitative differences in the performances of MFIN and MFINc for either loss function. Since, MFINc-SSIM provides the best quantitative results, we show interpolated images from this method and compare them against SCIN-SSIM. Both methods perform well when the motion between the neighbouring images is low. This is reflected in the absence of any structures in the error images in Fig. 4.1. However, RMSE is lower for SCIN because it produces a denoised interpolated image, while MFIN carries over the noise pattern from the neighbouring known image. Whenever there exists high motion between the images being interpolated, SCIN produces blurry images and often misses image structures. This can be observed in cases 2-4 in Fig. 4. For all these cases, MFINc (and also MFIN) produces sharp images and largely preserves structures in the images. Fig. 4.2 shows a case where MFINc additionally has a much better performance with respect to image alignment. Fig. 4.3 shows a representative case, with small improvement in image alignment, yet worse RMSE and SSIM values for MFINc. Finally, Fig. 4.4 shows a case, where MFINc produces worse alignment of structures than SCIN.

Fig. 4.4 shows a comparison of a representative motion field predicted by MFINc with that computed via the GS registration algorithm. We can see that the motion field produced by MFINc is smooth and has sharper motion boundaries. The reason for this might be that the used registration is more regularized due to its parametric model, where motion is defined by a grid of control points with 4x4 pixel spacing and linearly interpolated in between. This might also explain the higher error in evaluation of the flow field predicted by the network over the whole image (RefMotErrFl) than only over the liver (RefMotErrFlLiver).

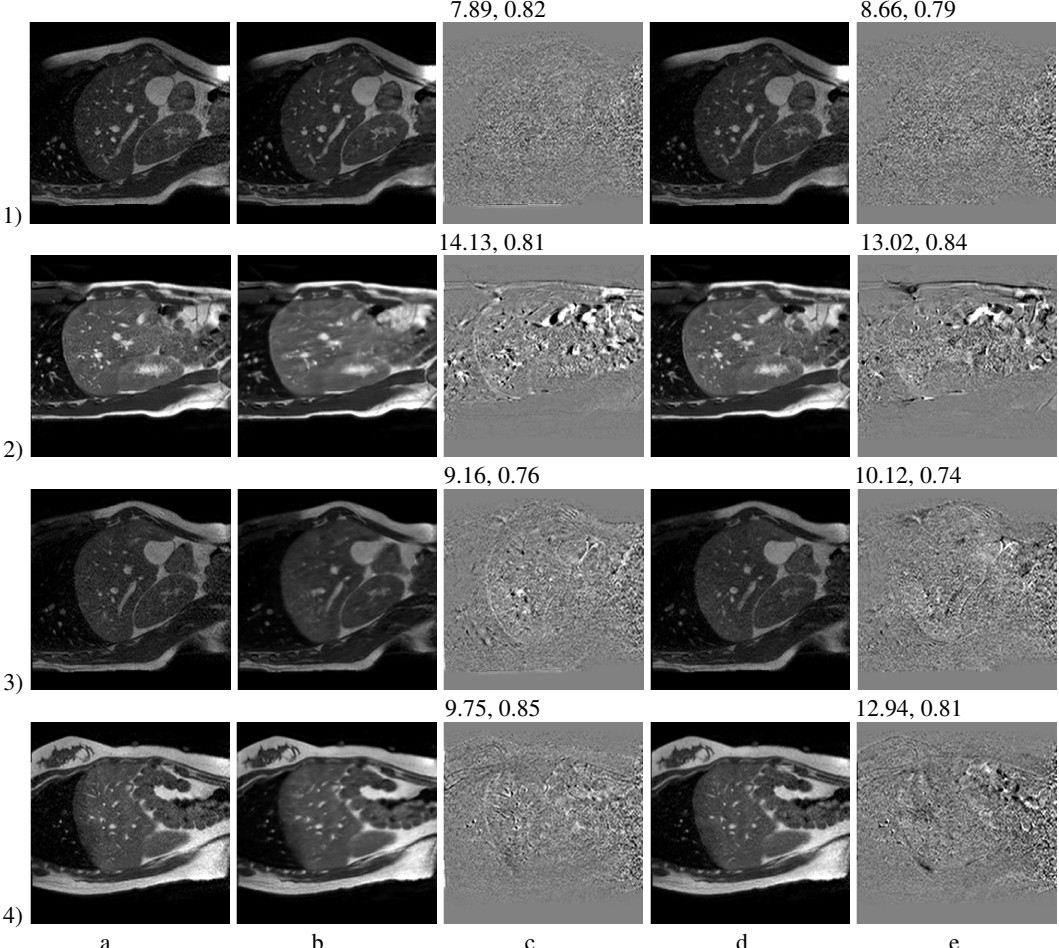

Figure 3: Ground truth images (a), SCIN-SSIM results (b) and difference images (c), MFINc-SSIM results (d) and difference images (e). Rows: (1) low motion case, (2)-(4) high motion cases, where MFINc produces (2) much better, (3) slightly better and (4) worse structure alignment that SCIN. (RMSE, SSIM) pairs are indicated over the respective errors images.

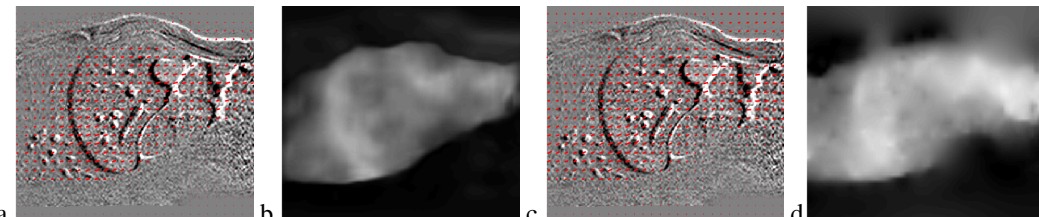

Figure 4: (a) Motion field $\mathbf{F}_{t \to t+1}$ from MFINc-SSIM overlaid on $N_{t+1}$-$N_t$ and (b) corresponding motion magnitude image. (c) $\mathbf{F}^{gs}_{t \to t+1}$ from gold standard registration and (d) its magnitude image.

## 5 Conclusion

In this article, we proposed a framework for temporal image interpolation that incorporates the prior knowledge that changes in the images over time are caused by the motion of the visible structures in the images. We showed the advantages of this approach over naive direct interpolation in the intensity space. Although we presented results in the setting of 4D MRI reconstruction, the method may be extended to other scenarios where the content of temporal sequences does not change over time.

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
