# OpenReview forum: "Temporal Interpolation via Motion Field Prediction"
_MIDL.amsterdam/2018/Conference — MIDL 2018 Oral_

### Review · AnonReviewer3 · 2018-05-10
**Novel and general technique for temporal interpolation; applied to 4D MR. The strongest paper in my stack, hands down.**

**Rating:** 4
**Confidence:** 2

**Review:**

Previous deep learning-based approaches to temporal interpolation avoid explicit motion field prediction, while other deep learning methods aim to predict only motion fields. This approach aims to predict motion fields as an intermediate step towards interpolation; moreover they are estimated in an unsupervised way. The technique is applied to 4D magnetic resonance (MR) imaging.

Strengths
- The method is novel and very well motivated
- Excellent, extensive lit review (from recent work on temporal forward prediction in video, to DL-based motion field estimation)
- Paper evaluates several variants of the techniques
- Shows qualitative improvements

Weaknesses
- Quantitative results are somewhat inconclusive; and difficult to interpret (Table 1) with all the various error metrics (the metrics are discussed in the text, but how does the reader infer what's important in the table)
- Some application-specific concepts, like the navigator slices, could be explained better (i.e. for someone who does not know much about MRI)
- No detail on timing

Questions/comments:

- Consider giving more detail on the navigators (even a diagram with navigator slices and data slides); I thought the rest of the paper was very clear but was lost by the first paragraph of the introduction.

- The VAE approach to bidirectional motion fields is mentioned and the comment is "it is unclear whether it would be able to faithfully interpolate breathing motion patterns in abdominal navigators"; why not try it and offer it as a baseline?

- It would be nice to re-position Figure 1 (if possible) to be on the same page as the beginning of Section 3, so one does not need to flip back and forth between the figure and the text explaining it

- Bi-directional motion field estimation is interesting and motivated by a "multi-task" like phenomenon; why not evaluate this in an ablation study? Same with the cycle consistency.

- The use of SSIM is also interesting; Eq 2 does not really like "L" (as the generic loss) to either L_2 or SSIM. Perhaps make this explicit. There's also an issue in Section 4.4 where L_2 loss and SSIM are substituted for "L_recon". But technically it is a substitution for "L" as L_recon has the forward and backward addition.




**Special Issue:**

Yes

---

### Review · AnonReviewer1 · 2018-05-10
**Efficient technique for temporal interpolation of 4D MRI Images with appropriate evaluation**

**Rating:** 4
**Confidence:** 2

**Review:**

The authors proposed a convolutional neural network (CNN) based methodology for temporal interpolation of 4d MRI by estimation the bi-directional motion fields, which as compared to available methods, also reduces the number of registrations require for 4D reconstruction, hence reducing time and computational complexity alongside improved performance.

Strengths
- The introduction and literature review is well written
- Evaluation methodology is appropriate.
- Paper evaluates several variants of the techniques
- Figure 1 shows clear qualitative improvements, however I suggest that this figure should be in the results section.

Weaknesses
-  Discussion is required to further elaborate the quantitative results
-  Addition of some reference on the domain specific terms would be an added advantage
-  Runtime analysis of the proposed methodology has not been discussed in detail, although it is mentioned that the proposed algorithm is computationally efficient than other techniques


**Special Issue:**

Yes

---

### Review · AnonReviewer2 · 2018-05-14
**4D image interpolation by symmetric motion field prediction**

**Rating:** 3
**Confidence:** 3

**Review:**

The proposed method implements a regression network that predicts a transformation map and subsequently resamples the data. It does so with a joint loss on image similarity and displacement field smoothness. It also adds a symmetric term on the transformation to ensure that forward and backward transformation is consistent. Methodologically the paper is novel, well written.

The paper is very extensively validated, provides different metrics measuring various characteristics of the resampled image and deformations, and provides useful qualitative visualisations of the proposed algorithm’s performance. Results of table 1 are non-surprising, as the SCIN network optimising for L2 performs better on MSE, and the one optimising for SSIM performs better for SSIM. The proposed model does indeed perform better with other unbiased metrics. One big flaw of the proposed work is that it is not validated against classic image registration techniques which have been used for this application for many years. Further discussion of results would also have helped.

It is unfortunate that the resampling part of the model is not described in detail, as it would be interesting to understand how the differentiation works. Sections 1-2 and Fig2 describe what it does not do, but I fail to understand how the interpolation subnetwork actually work? Is it only bilinear resampling of a deformation field? If so, why does the subnet of Fig2 (top right) look like a standard feed forward neural network?

It would also been interesting to better understand the need for predicting the deformation field as an intermediate result beyond performance. Is the deformation really field necessary? How often are the assumptions broken (no out-of-plane movement, similar signal, etc)? Are occlusions/sliding an issue (I would argue so due to lung motion and sliding)? Some of assumptions of the paper would rarely hold, so it would be nice to better understand these model limitations.

**Special Issue:**

No

---

### Comment · ~Bram_van_Ginneken1 · 2018-05-18
**Selection for longlist for special issue Medical Image Analysis**

Dear authors,

Congratulations on your acceptance to MIDL! We have selected your paper on the longlist for the Medical Image Analysis Special Issue. Please read this page:
https://midl.amsterdam/special-issue-in-medical-image-analysis/
Please answer the three questions that are listed on that page about your interest in submitting to the special issue, potential overlap with other publications, and related publications.

You can post your answer here directly below on openreview.net, or mail me directly at bram.vanginneken@radboudumc.nl.

Best regards, Bram

---

### Decision · Program_Chairs · 2018-05-15
**Paper89 Acceptance Decision**

Oral